# Hydrochemical Characteristics and Hydrogeochemical Simulation Research of Groundwater in the Guohe River Basin (Henan Section)

**Furong Yu** [1,2] **, Dongxu Zhou** [1] **, Zhiping Li** [1,2,]***** **and Xiao Li** [3]

[1] College of Geosciences and Engineering, North China University of Water Resources and Electric Power, Zhengzhou 450046, China; yufurong@ncwu.edu.cn (F.Y.); 201604217@stu.ncwu.edu.cn (D.Z.)
[2] Collaborative Innovation Center for Efficient Utilization of Water Resources, Zhengzhou 450046, China
[3] Sinopec North China Oil and Gas Branch, Yulin 719000, China; lix202204@126.com
***** Correspondence: lizhiping@ncwu.edu.cn

**Abstract:** With the implementation of the policy of ecological protection and high-quality development of the Yellow River Basin, the Guohe River Basin, which is close to the middle reaches of the Yellow River Basin, is also an important part of future development. Mathematical statistics, the Piper diagram, ion proportion coefficient method, Gibbs diagram and reverse hydrogeochemical simulation are used to analyze the chemical characteristics and evolution of groundwater in the Guohe River Basin (Henan Section). The dominant ions in the study area are $HCO_3^-$ and $Na^+$, and the three-layer aquifer has obvious zoning characteristics. The results show that the chemical types of shallow groundwater in this area are $HCO_3-Na \bullet Mg \bullet Ca$, intermediate $HCO_3$-Na and deep $HCO_3-Na$. Using the ion proportion coefficient method, it is found that $Na^+$, $Ca^{2+,}$ and $Mg^{2+}$ in the groundwater aquifer undergo cation exchange in the aquifer. According to the reverse hydrogeochemical simulation, gypsum in the three aquifers is in a dissolved state, carbonate and sulfide ores in the shallow layer are dissolved, dolomite and halite in the intermediate layer are dissolved, calcite and sulfide ores are precipitated and carbonate, halite and sulfide ores in the deep layer are precipitated; the hydrogeochemical evolution model is established to find that $Ca^{2+}$ in groundwater displaces $Na^+$ in the aqueous medium. This research can provide a scientific basis for the rational development and utilization of groundwater and ecological protection in the Yellow River Basin.

**Keywords:** hydrochemistry; inverse hydrogeochemical simulation; evolution model; Guohe River Basin (Henan Section)





## 1. Introduction

The chemical composition of groundwater is mainly affected by geological conditions, the extent of chemical weathering of rocks, characteristics of aquifers and human activities [1,2]. Its formation and evolution are the long-term result of groundwater diagenesis. With the circulation of groundwater, the process is more complex and changeable [3]. The quality of groundwater in an area depends on the chemical composition of rock stratum and make-up water, the interaction between soil and water, the rock present in unsaturated areas and its interaction with make-up water, the recharge time of the aquifer and the action of the aquifer itself [4,5]. The definition of hydrogeochemistry was proposed in 1979 as a science with which to study various chemical interactions and physical phenomena between the hydrosphere and its surrounding natural environment [6]. In 1993, Shen Zhaoli et al. [7] compiled the basis of hydrogeochemistry and established the discipline system of hydrogeochemistry.

The relevant information of groundwater circulation characteristics from different angles can be acquired by studying the chemical composition, and distribution characteristics of groundwater can be obtained [8,9]. Lloyd, S. Adams [5,10] and others used a variety of

conventional statistical methods (such as mathematical statistical analysis, factor analysis, etc.) to conduct mathematical statistical analysis and interpretation. Hitchon et al. [11], Gupta et al. [12] and S. Kumar [13] added multivariate analysis to the interpretation of groundwater chemistry. After decades of development, hydrogeochemical theory has been continuously improved, and research methods such as ion proportion coefficient method and water rock interaction type have been continuously enriched [14,15]. Rabelani M. [16] studied the hydrogeochemical characteristics of groundwater in Grand Giani City, Limpopo Province, South Africa using a Piper diagram, Gibbs diagram and ion ratio. Kaur L. et al. [17] investigated the chemical composition of groundwater in the Panipa special district (semi-arid alluvial area) of Haryana, India by using a Piper diagram and ion proportion relationship. N.M. Refat Nasher [18] used factor analysis and ion ratio to analyze the hydrogeochemical characteristics of groundwater in three basins in Bangladesh. The depth of ion ratio analysis is adopted to describe the interpretation of calcite and dolomite to the traditional hydrogeochemical concept [19]. Remy R. [20] used groundwater and surface water samples collected in different seasons to explain the hydrogeochemical process of Kattumannarkoil Taluk. Aher S. et al. [21] explored the hydrogeochemical characteristics of the Pravara River Basin using groundwater samples collected on the cross-section of the basin. Accurate assessment of groundwater hydrogeochemical characteristics and groundwater quality is helpful and necessary for sustainable groundwater protection and management [22,23].

In recent years, scholars have used isotope analysis and hydrogeochemical simulation methods to study hydrochemical characteristics to good effect [24–26]. Ren et al. [27] found that groundwater in any area has a unique chemical composition, which is caused by many processes such as the interaction between water and rock during recharge and groundwater flow through the aquifer and the long-term water storage of the aquifer. Wei et al. [28], Subba Rao et al. [29] and Wu et al. [30,31] aimed to delineate the process responsible for the geochemical changes of aquifer water environment. Prasanna et al. [32] used the method of digital system analysis to calculate the saturation index of groundwater. Parkhurst et al. [33] began the classification of mineral saturation index in groundwater samples using PHREEQC in 2013.

Through hydrochemical analysis of the shallow groundwater in the Guohe River Basin (Anhui section), Ma Tao et al. [34] found that the shallow groundwater in the study area is susceptible to human activities and atmospheric rainfall. Zheng Tao et al. [35] investigated the chemical characteristics and genetic mechanism of groundwater in the central area of the Guohe River Basin. They found that there are significant vertical differences in the chemical composition of groundwater, and the formation of groundwater hydrochemical characteristics is affected by water–rock interaction, cation exchange and human activities. At present, there are few studies on hydrochemistry that have focused on the upper reaches of the Guohe River Basin. In the context of ecological protection and high-quality development in the Yellow River Basin, the research aims to determine the chemical characteristics of shallow, mid-deep and deep groundwater in the Guohe River Basin (Henan Section), and to identify its salient hydrogeochemical formation, assess the transport of groundwater components through reverse hydrogeochemical simulation, quantitatively study the chemical components of groundwater using the hydrogeochemical evolution model and reveal the chemical formation process of groundwater and the spatial transport trends in its groundwater hydrochemistry. The results may provide a scientific basis for rational development and use of groundwater and ecological protection therein.

## 2. Materials and Methods

### 2.1. Regional General Situation

The study area is located in the east of Henan Province, in the upper reaches of the Guohe River Basin, and forms an important part of the Huaihe River Basin, covering an area of about 14,000 km$^2$, with geographic coordinates of 113°22′–115°39′ E and 33°22′–34°50′ N. The Guohe River Basin is located in the warm temperate zone and has obvious monsoonal

climate characteristics. According to the precipitation data from meteorological stations in Kaifeng, Shangqiu and other places, the annual average precipitation over the basin is 638 to 690 mm, and the trend of precipitation increases slightly toward the southeast, consistent with the direction of the Guohe River. The humidity is the highest from June to September, and the annual average temperature is 13 to 15 °C. The evaporation intensity decreases from west to east, and the evaporation is the strongest from May to June. The lithology of the Pre-Cenozoic Middle Jixian System is mainly metamorphic sandstone and schist. The Cambrian lithology is mainly limestone and dolomite, with a thickness of 21.2 to 218.29 m. The Ordovician-Cambrian is mainly composed of limestone and dolomite with a total thickness of more than 400 m. The main lithology of the Cenozoic Paleogene is claystone, sandy claystone, etc., and sand layers are distributed alternately therein. In some areas, there are sand-argillaceous limestones, and the strata are in angular unconformity contact with the Cambrian and Ordovician strata.

### 2.2. Sample Collection and Research Methods

The project team collected 180 sets of groundwater samples in the study area from September to October 2020, including 108 sets of shallow groundwater, 34 sets of mid-deep groundwater and 38 sets of deep groundwater (Figure 1). The water samples were sent to the Experimental Testing Center of the Natural Resources Monitoring Institute of Henan Province for testing. The sampling data analysis included seven constant inorganic components ($K^+$, $Na^+$, $Ca^{2+}$, $Mg^{2+}$, $Cl^-$, $SO_4^{2-}$, and $HCO_3^-$), as well as pH, water temperature, COD, total hardness and TDS. The assay for $Na^+$ relied upon atomic absorption spectrophotometry. The tests for $Ca^{2+}$, $Mg^{2+}$, $Cl^-$ and $SO_4^{2-}$ relied upon ion chromatography; $HCO_3^-$ was determined by on-site titration; WTW portable multi parameter water quality analyzers were used to measure the TDS, water temperature and pH value of spring water. The measurement accuracy is 1 mg/L, 0.1 °C and 0.01 pH units, respectively; COD was measured by the three-position fluorescence method.

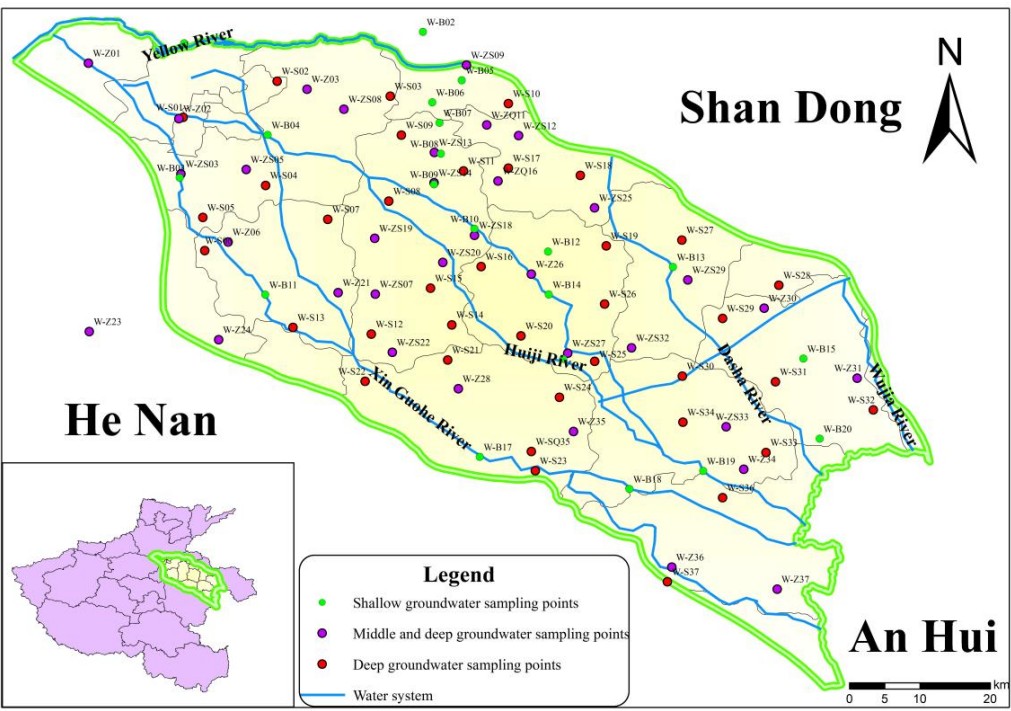

**Figure 1.** Distribution of hydrochemical sampling points in the Guohe River Basin.

The geological data of the study area show that the boundary between shallow and middle deep groundwater in the Guohe River Basin is set at about 50 m, at 50 to 300 m the middle aquifer (the Kaifeng control depth is about 400 m) is found, and the deep

control depth is between 300 and 600 m. Shallow groundwater samples come from rural decentralized water supply wells and farmland irrigation wells, with a depth of 8 to 50 m; the intermediate groundwater comes from rural centralized water supply wells with a depth of 50 to 300 m; the deep groundwater comes from the water supply well serving the urban waterworks, with a depth of more than 300 m.

Mathematical statistics could be used to study the variational characteristics of the coefficient of variation and salinity of the three-layer aquifers in the study area. The main source of ions was explored by using a Gibbs diagram to determine the hydrolithic effect of ions in groundwater. Depending on the water chemistry, TDS partition and hydrogeological conditions, using PHREEQC software (Parkhurst and others, USDS), the research calculated mineral saturation index (SI) using the reverse hydrogeochemical simulation, simulated the shallow, intermediate, and deep groundwater migration paths, traced dissolved mineral phases as they underwent precipitation and mapped the Guohe River Basin along the flow profile on the hydrogeochemical evolution model, thus revealing the hydrogeochemical formation process of groundwater in the South of the Guohe River Basin.

### 2.3. Statistical Analysis

Descriptive statistics provide a general view of the hydrochemical characteristics [36]. To prevent subjective and objective errors in the test, the cation anion balance test method is adopted here, and the test formula is determined as follows:

$$E = \frac{\Sigma m_c - \Sigma m_a}{\Sigma m_c + \Sigma m_a} \times 100\% \tag{1}$$

where $E$ denotes the relative error, %; $m_c$ and $m_a$ are the milligram equivalent concentrations of cations and anions, meq/L.

If the relative error is $\pm 5\%$, it indicates that the data are valid and reliable. Substituting the measured data into Formula (1), the collected 108 groups of shallow groundwater, 34 groups of intermediate groundwater and 38 groups of deep groundwater meet the relative error criterion of $\pm 5\%$; that is, they satisfy the prerequisite conditions.

### 3. Results and Discussion

#### 3.1. Analysis of Groundwater Chemical Types

3.1.1. Statistical Characteristics of Main Indicators: Analysis of Hydrochemical Types and Characteristics

A mathematical statistical analysis was conducted on the groundwater sample data in was performed (Table 1), and the average content distribution diagram of major ions and indicators was drawn (Figure 2). The coefficient of variation (C.V.) can represent the dispersion degree of various indexes in groundwater and the complexity of influencing factors of groundwater chemical composition and its evolution [37]. The coefficient of variation of $K^+$, $Cl^-$ and $SO_4^{2-}$ in the shallow layer is large, indicating that the shallow layer is strongly affected by human activities. It can be seen that the coefficient of variation of $Na^+$, $Ca^{2+}$ and $Mg^{2+}$ in the middle and deep layer is large. This phenomenon may be affected by the recharge of shallow groundwater or cation exchange adsorption. From shallow to intermediate to deep, the groundwater salinity in the study area presents a trend of high to low to low. This zonation indicates that the vertical component difference of the three aquifers may be caused by the surrounding rock environment and the intensity of water–rock interaction. Meanwhile, the salinity of the shallow layer is generally higher than that of the intermediate layer, which is possibly influenced by human activities and farmland irrigation. The results of this study differ from those of Zheng Tao et al. [35], wherein an analysis of the low → high → low trend of groundwater stratification mineralization in The Guohe River Basin (Anhui Section) is revealed.

**Table 1.** Concentration and content statistics of main hydrochemical indicators (unit: mg/L).

| Type | Project | K$^+$ | Na$^+$ | Ca$^{2+}$ | Mg$^{2+}$ | Cl$^-$ | SO$_4{}^{2-}$ | HCO$_3{}^-$ | Hardness | TDS | TDS [35] |
|---|---|---|---|---|---|---|---|---|---|---|---|
| Shallow | Mean | 2.55 | 147.66 | 89.77 | 73.84 | 117.25 | 126.50 | 686.91 | 528.12 | 917 | 823 |
| | S.D. | 4.79 | 158.94 | 49.31 | 32.68 | 122.55 | 159.28 | 191.52 | 225.39 | 507 | 321 |
| | Min. | 0.30 | 15.90 | 15.30 | 24.90 | 12.30 | 3.82 | 359.00 | 141 | 400 | 289 |
| | Max. | 41.80 | 1420.00 | 384.00 | 205.00 | 696.00 | 1287.00 | 1640.00 | 1780 | 4372 | 1620 |
| | C.V. [1] | 1.88 | 1.08 | 0.55 | 0.44 | 1.05 | 1.26 | 0.28 | 0.43 | 0.55 | 0.39 |
| Intermediate | Mean | 1.58 | 203.01 | 18.18 | 16.26 | 77.86 | 76.28 | 398.59 | 112.35 | 622 | 1322 |
| | S.D. | 0.37 | 71.45 | 21.08 | 18.66 | 65.99 | 52.01 | 114.69 | 126.74 | 178 | 264 |
| | Min. | 0.60 | 37.00 | 3.30 | 1.70 | 21.50 | 21.50 | 251.00 | 15 | 366 | 869 |
| | Max. | 2.40 | 353.00 | 87.80 | 88.90 | 320.00 | 211.00 | 807.00 | 585 | 1080 | 1710 |
| | C.V. [1] | 0.24 | 0.35 | 1.16 | 1.15 | 0.85 | 0.68 | 0.29 | 1.13 | 0.29 | 0.20 |
| Deep | Mean | 11.16 | 8.19 | 60.80 | 64.73 | 372.61 | 61.63 | 560.00 | 1.56 | 200 | 792 |
| | S.D. | 15.10 | 10.13 | 42.35 | 40.39 | 60.51 | 78.82 | 136.33 | 0.33 | 58 | 139 |
| | Min. | 1.00 | 55.10 | 3.00 | 1.40 | 18.80 | 22.10 | 244.00 | 13 | 347 | 561 |
| | Max. | 2.40 | 330.00 | 94.60 | 57.90 | 188.00 | 193.00 | 490.00 | 475 | 950 | 970 |
| | C.V. [1] | 1.35 | 1.24 | 0.70 | 0.62 | 0.16 | 1.28 | 0.24 | 0.21 | 0.29 | 0.18 |

[1] (The coefficient of variation is dimensionless).

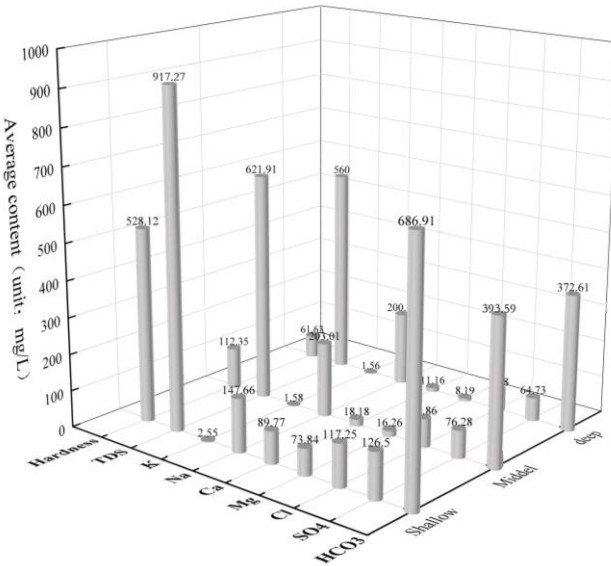

**Figure 2.** Average content distribution of main ions and indicators in the three-layer aquifer.

### 3.1.2. Analysis of Hydrochemical Types and Characteristics

At present, many scholars use Piper three-line diagrams to analyze the ion composition and variation characteristics of surface water and groundwater [38,39]. D. Gamvroula and D et al. [40] used this method to explain in detail the distribution and characteristics of macronutrient ions in the groundwater of Megara Basin. Piper tri-maps of shallow, intermediate, and deep layers in the study area were drawn (Figure 3). The results showed that Na$^+$, Ca$^{2+}$ and Mg$^{2+}$ ions are abundant and evenly distributed in shallow groundwater, and the hydrochemical types are mainly HCO$_3$-Na•Mg•Ca, accounting for 35.19% overall. The hydrochemical types of some sampling sites are Cl•HCO$_3$•SO$_4$-Na•Mg and HCO$_3$-Ca•Mg, accounting for 14.82% overall. HCO$_3{}^-$ is abundant in the intermediate groundwater, and the main cations are Na$^+$. The hydrochemical type of HCO$_3$-Na is 44.12% overall. Some are of the HCO$_3$•Cl•SO$_4$-Na type, accounting for 17.65% overall, and the proportion of HCO$_3$•Cl-Na type is 14.71%. The type of deep groundwater is similar to that of middle deep groundwater, mainly HCO$_3$-Na type, accounting for 60.53% overall with some HCO$_3$•Cl-Na type present, accounting for 23.68% overall.

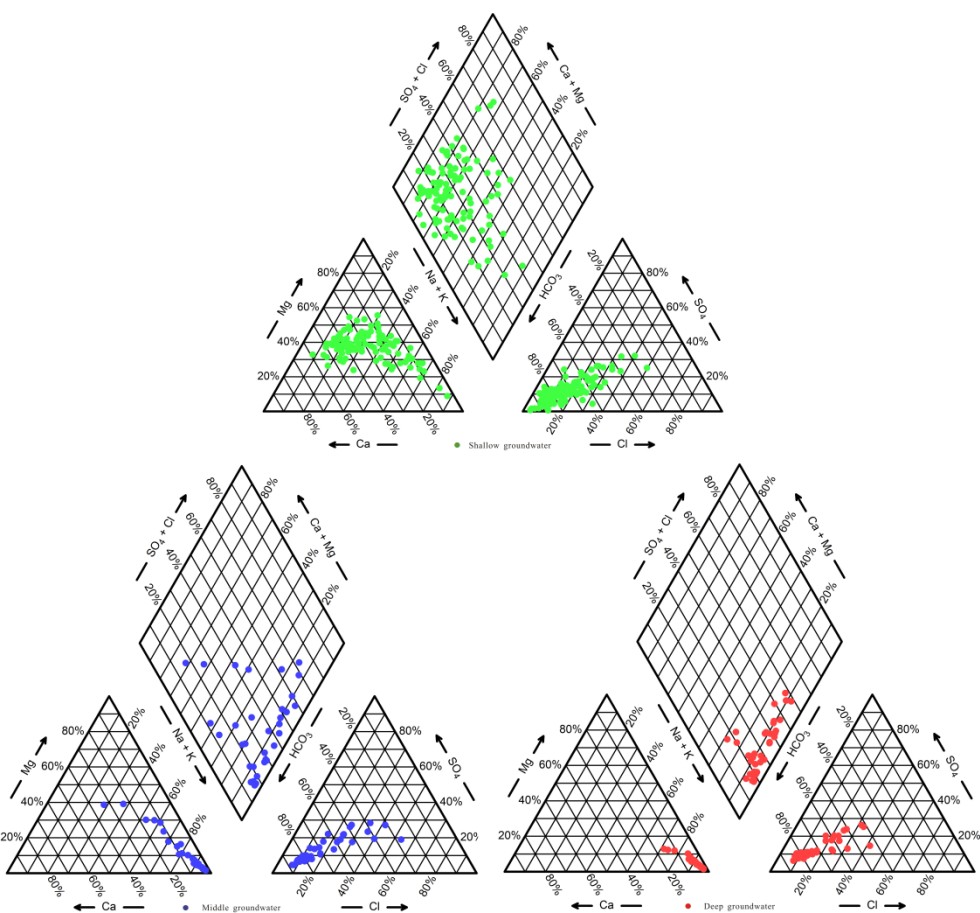

**Figure 3.** Piper tri-maps: shallow, intermediate, and deep groundwater in the Guohe River Basin (Henan Section).

Along the direction of groundwater flow, the order of cation contents in shallow groundwater in the upper reaches of the study area is $Ca^{2+} > Mg^{2+} > Na^+$, the order of cation contents in the lower reaches is $Na^+ > Mg^{2+} > Ca^{2+}$, and the order of anions is $HCO_3^- > SO_4^{2-} > Cl^-$. The order of anionic contents in the intermediate groundwater is $HCO_3^- > Cl^- > SO_4^{2-}$, and the order of cations is $Na^+ > Ca^{2+} > Mg^{2+}$. The order of cations in the deep groundwater is different from that in the other two layers, so the exchange effect of the deep groundwater is weak. Meanwhile, $HCO_3^-$ and $Na^+$ are found to play a dominant role in groundwater with increasing depth, and their basic hydrochemical type is characterized by the transformation from $HCO_3$-Na•Mg•Ca type in shallow groundwater to $HCO_3$-Na type at intermediate depth, and then to $HCO_3$-Na type in deep groundwater, with zoning characteristics.

### 3.2. Analysis of Water-Rock Interaction

3.2.1. Ion Proportion Relation and Component Origin Analysis

$\gamma Na^+/\gamma Cl^-$ is known as groundwater genetic coefficient, which can be used to reflect the sources of $Na^+$ and $Cl^-$ in groundwater [7]. The leachate value of halite-bearing strata is about 1 [41]. According to the proportion coefficient of water ions in the study area, Figure 4a shows that the ratio of $\gamma Na^+/\gamma Cl^-$ in groundwater is mostly greater than 1, indicating that $Na^+$ is not completely derived from the dissolution of halite, but also the dissolution of other sodium-bearing ores. Meanwhile, it is found that $\gamma Na^+/\gamma Cl^-$ is present as follows: deep > intermediate > shallow layers. This indicates that the deep and intermediate layers are relatively closed. It can be seen in Figure 4b that $HCO_3^-$ is not completely derived from the dissolution of calcite. The shallow, intermediate and deep

underground water samples in the study area are distributed on the side of the isoratio line that is biased towards $HCO_3^-$, indicating that the alternating adsorption of $Na^+$ and $Ca^{2+}$ cations or desulfuration may occur in the study area, which reduces $Ca^{2+}$ and increases $HCO_3^-$. Figure 4c indicates that the shallow groundwater and intermediate groundwater are distributed on both sides of the contour line, indicating that gypsum leaching may have occurred. The deep groundwater is all below the contour line, biased to the side of $SO_4^{2-}$, suggesting that cation alternating adsorption may have occurred to decrease $Ca^{2+}$ and increase $Na^+$ content. The deep groundwater is all below the isoratio line, which is on the side of $SO_4^{2-}$. In addition to the cation alternation adsorption, which can reduce $Ca^{2+}$, the dissolution of other sulfate minerals such as mirabilite, or even the contribution of sulfide oxidation to $SO_4^{2-}$ content cannot be ruled out. Figure 4d shows a positive correlation between $Ca^{2+}+Mg^{2+}$ and $HCO_3^-$ in the shallow layer, indicating calcite and dolomite solubilization. Some of the shallow groundwater is distributed on one side of $Ca^{2+}+Mg^{2+}$, suggesting that the solubilization of other calcium and magnesium minerals, such as gypsum and tremolite, may have occurred. The groundwater in the intermediate layer is inclined to $HCO_3^-$, implying that the alternate adsorption of cations may occur in the study area. According to Figure 4e, water samples from intermediate layers in the study area have $\gamma (Ca^{2+}+Mg^{2+})/\gamma (HCO_3^- + SO_4^{2-})$ ratios of less than 1, indicating that the main sources of $Ca^{2+}$ and $Mg^{2+}$ are the dissolution of evaporite and silicate minerals, suggesting that the negative charges of $HCO_3^-$ and $SO_4^{2-}$ are balanced by another cation such as $Na^+$. $HCO_3^-$ and $SO_4^{2-}$ may be controlled by a series of hydrochemical actions such as silicate hydrolysis, sulfide oxidation and cation-alternating adsorption, especially in intermediate and deep groundwater. The obvious enrichment characteristics of $HCO_3^-$ and $SO_4^{2-}$ warrant further discussion. Figure 4d and e show that there is significant enrichment of $Ca^{2+}$ and $Mg^{2+}$ in the shallow layer. This enrichment is probably affected by human activities, but the enrichment of $Ca^{2+}$ and $Mg^{2+}$ in the shallow layer caused by the evaporation concentration cannot be ruled out, which also warrants further discussion.

Through $[\gamma(Ca^{2+}+Mg^{2+})-\gamma(SO_4^{2-}+HCO_3^-)]/\gamma(Na^+-Cl^-)$ ratio analysis of the cation exchange, adsorption is found to occur; $\gamma(Na^+-Cl^-)$ represents the change of $Na^+$ content caused by other sodium-bearing minerals except halite dissolution. $[\gamma(Ca^{2+}+Mg^{2+})-\gamma(SO_4^{2-}+HCO_3^-)]$ represents the changes of $Ca^{2+}$ and $Mg^{2+}$ contents caused by other substances other than gypsum, dolomite, and calcite. As shown in Figure 4f, the slope in shallow groundwater is $-0.925$, the slope in intermediate groundwater is $-1.032$, and the slope in deep groundwater is $-0.984$ (all close to $-1$), indicating that the change is linear, and that cation exchange adsorption has occurred at the spatial scale of groundwater.

### 3.2.2. Gibbs Diagrams and Their Implications

The chemical natural formation mechanism of groundwater mainly includes atmospheric precipitation, water-rock interaction and evaporation concentration [42,43]. The Gibbs diagrams of shallow, intermediate and deep groundwater in the study area demonstrate that the shallow groundwater body in the study area is jointly affected by water–rock interaction and evaporation (albeit mainly by water–rock interaction); the sampling points in the intermediate and deep groundwater are situated in the middle area. The chemistry of intermediate groundwater is controlled by water–rock interaction; that is, the formation mechanism of groundwater runs from shallow to deep, and weathering of the rock is continuously intensified (Figure 5).

### 3.3. Reverse Hydrogeochemical Simulation

In the reverse hydrogeochemical simulation, the selection of appropriate simulation paths and possible mineral phases is the key to the establishment of the model, so as to determine the main SI and the transformation amount of each mineral phase on the simulation path, so as to study the hydrogeochemical formation [44]. According to the hydrogeological conditions prevailing across the study area [45–47], calcite ($CaCO_3$), dolomite

(CaMg (CO$_3$)$_2$), gypsum (CaSO$_4$•2H$_2$O), halite (NaCl), CO$_2$, O$_2$, H$_2$S, NaX and CaX$_2$ were selected as the reaction mineral phases.

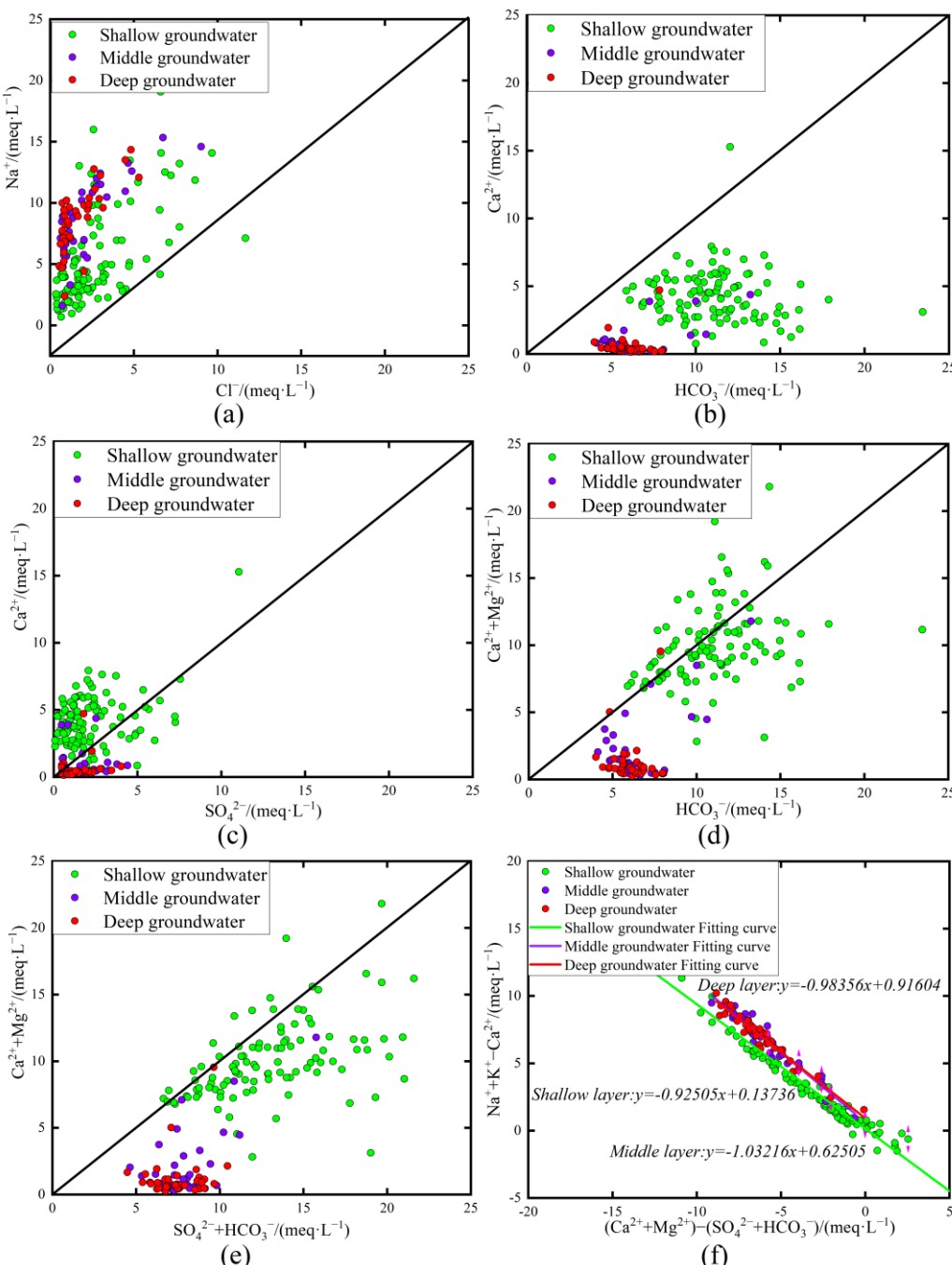

**Figure 4.** Ion proportion coefficients of shallow, intermediate, and deep groundwater in the study area.

### 3.3.1. Mineral Saturation Index

The content of CO$_2$ is used as a proxy for the degree of dissolution of carbonate; H$_2$S and O$_2$ can distinguish the degree of dissolution of pyrite, tremolite and other sulfur-bearing minerals; according to the Gibbs diagram analysis results, calcite and dolomite were added as the main reactive mineral phases. NaX and CaX$_2$ are added according to results of the ion proportion coefficient method. The mineral SI indicates the saturation state of minerals relative to groundwater. The SI values at some representative points in the study area are listed in Table 2.

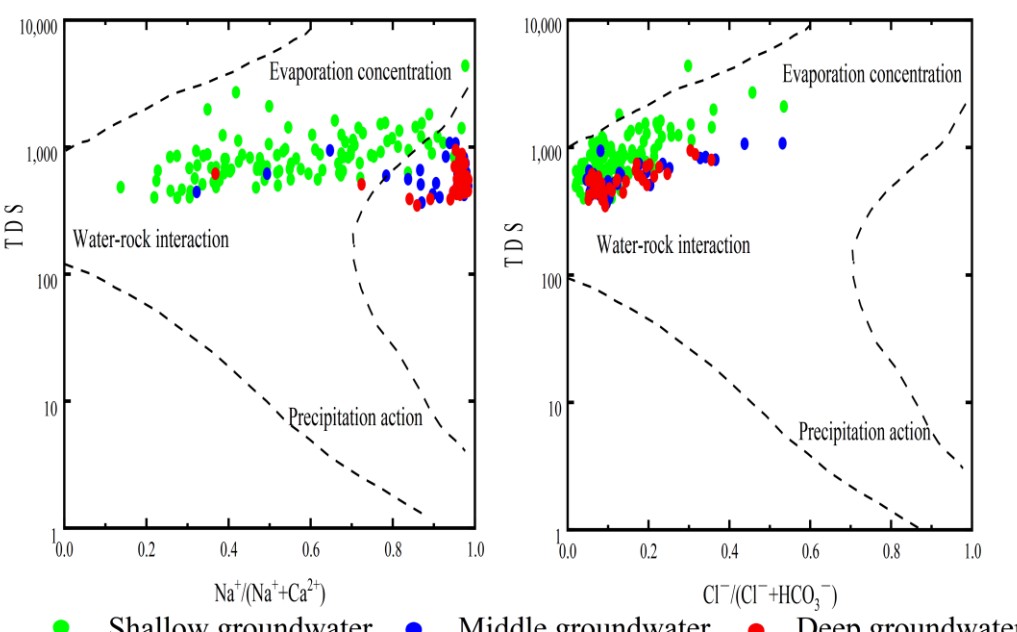

**Figure 5.** Gibbs diagram of shallow, intermediate and deep groundwater in the study area.

**Table 2.** Mineral SI of representative points of shallow, intermediate and deep layers in Guohe River Basin (Henan Section).

| Point | SI$_{Calcite}$ | SI$_{Dolomite}$ | SI$_{Gypsum}$ | SI$_{Halite}$ | Point | SI$_{Calcite}$ | SI$_{Dolomite}$ | SI$_{Gypsum}$ | SI$_{Halite}$ |
|-------|-------|-------|-------|-------|-------|-------|-------|-------|-------|
| W-Q06 | 0.38 | 0.60 | −2.44 | −7.62 | W-S08 | −0.08 | −0.07 | −3.30 | −6.91 |
| W-Q51 | 0.41 | 1.60 | −1.40 | −4.70 | W-S16 | 0.11 | 0.40 | −3.12 | −6.71 |
| W-Q88 | 0.54 | 0.88 | −2.01 | −8.01 | W-S22 | 0.45 | 1.14 | −2.96 | −6.73 |
| W-Z02 | −0.34 | −0.51 | −2.91 | −6.91 | W-S24 | 0.20 | 0.61 | −3.22 | −6.78 |
| W-Z35 | −1.13 | −2.18 | −3.39 | −7.01 | W-S30 | 0.19 | 0.54 | −3.25 | −6.81 |
| W-Z36 | −0.56 | −0.85 | −2.04 | −5.57 |  |  |  |  |  |

When SI > 0, it indicates that the mineral facies is in a supersaturated state and tends to be precipitated in the groundwater aquifer (otherwise, it will be further dissolved). The results show that the groundwater aquifers in the Guohe River Basin (Henan Section) are divided into distinct zones, calcite and dolomite in shallow and deep groundwater is probably precipitated, and calcite and dolomite in intermediate aquifers may be further dissolved in groundwater. Gypsum and halite always maintain their ability to dissolve further at the vertical spatial scale of the groundwater profile.

### 3.3.2. Mineral Saturation Index

According to the TDS, isoline is the main factor in the selection of reverse hydrogeochemical simulation paths; the shallow, intermediate and deep simulation paths in the study area are shown in Figure 6.

The measured pH, water temperature and major ion components (Table 3) are applied to PHREEQC for reverse hydrogeochemical simulation, and the results are summarized in Table 4.

The shallow simulation path (W-Q06→W-Q51) mainly involves the dissolution of calcite, dolomite, gypsum and sulfur-bearing minerals. In the simulation path for the intermediate layers, gypsum, salt and dolomite dissolution and precipitation of calcite and sulfide ores are the main factors influencing the hydrogeochemistry. From the simulation starting point (W-Z02) to its end (W-Z36), the contents of Na$^+$, Cl$^-$ and SO$_4^{2-}$ increase significantly; the contents of Ca$^{2+}$ and Mg$^{2+}$ rise slightly, and the contents of other ions change little. The dissolution of gypsum and the precipitation of calcite, dolomite, halite and

sulfide ores occur in the deep simulation path. The contents of Na$^+$, HCO$_3^-$ and SO$_4^{2-}$ in the water grow from the simulated starting point (W-S08) to its end (W-S30). The dissolution of NaX and precipitation of CaX$_2$ in the three-layer groundwater simulation path reveal that the cation exchange reaction between Ca$^{2+}$ and Na$^+$ is common in the aquifer, which also verifies the conclusion obtained by the ion proportional coefficient method. From the vertical space, the hydrochemical type changes from shallow HCO$_3$-Na•Mg•Ca type to intermediate HCO$_3$-Na type, and then to the deep groundwater dominated by almost all HCO$_3$-Na type; Na$^+$ and HCO$_3^-$ gradually become the dominant ions in groundwater. The proportions of Ca$^{2+}$, Mg$^{2+}$, Cl$^-$ and SO$_4^{2-}$ in the water decrease with increasing depth. The hydrogeochemical formation of groundwater in the Guohe River Basin (south section of the River) is quantitatively revealed by the amount of mineral dissolved deposits calculated by reverse hydrogeochemical simulation.

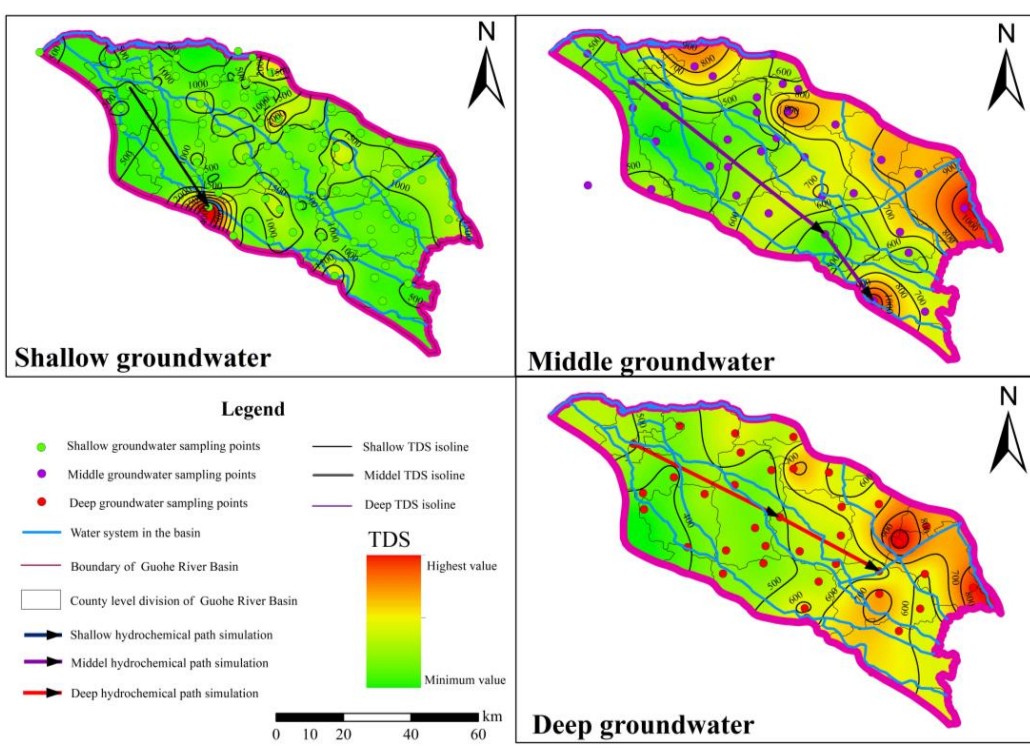

**Figure 6.** Shallow, intermediate and deep simulation paths of Guohe River Basin (Henan Section).

**Table 3.** Ion table of constant components at reverse hydrogeochemical simulation point (unit: mg/L).

| Point | K$^+$ | Na$^+$ | Ca$^{2+}$ | Mg$^{2+}$ | Cl$^-$ | SO$_4^{2-}$ | HCO$_3^-$ | pH | Temp. | TDS |
|---|---|---|---|---|---|---|---|---|---|---|
| W-Q06 | 2.30 | 32.20 | 92.30 | 33.40 | 27.90 | 13.10 | 472.00 | 7.27 | 19.0 | 453 |
| W-Q51 | 2.40 | 1420.00 | 34.00 | 114.00 | 696 | 1287.00 | 1640.00 | 7.57 | 18.7 | 4372 |
| W-Z02 | 1.80 | 130.00 | 12.20 | 9.60 | 34.00 | 24.30 | 294.00 | 7.57 | 19.5 | 403 |
| W-Z35 | 1.10 | 164.00 | 4.40 | 3.10 | 21.50 | 21.50 | 359.00 | 7.18 | 16.6 | 420 |
| W-Z36 | 1.75 | 336.00 | 22.00 | 21.70 | 320.00 | 154.00 | 282.00 | 7.21 | 20.1 | 1080 |
| W-S08 | 1.60 | 176.00 | 4.40 | 2.50 | 25.50 | 28.90 | 344.00 | 8.18 | 23.5 | 438 |
| W-S16 | 1.30 | 190.00 | 5.00 | 3.00 | 38.60 | 41.20 | 335.00 | 8.28 | 28.6 | 477 |
| W-S30 | 1.20 | 230.00 | 3.70 | 2.20 | 25.40 | 45.50 | 461.00 | 8.43 | 26.4 | 552 |

### 3.3.3. Analysis of the Hydrogeochemical Evolution Model

The hydrogeochemical evolution model on spatial scale was developed by combining geological and hydrogeological conditions, hydrogeochemical processes and main anions in groundwater at the time of groundwater hydration. To reveal the horizontal transport of anions and cations along the groundwater flow path, and through the changes in their relative concentrations with depth, the changes of main anions and cations in the

vertical spatial scale are analyzed, and the spatial transport associated with the prevailing groundwater hydrochemistry is studied (Figure 7).

**Table 4.** Reverse hydrogeochemical simulation results of Guohe River Basin (Henan Section) (unit: mmol/L).

| Mineral Phase | Shallow Simulation Path | Intermediate Simulation Path | | Deep Simulation Path | |
|---|---|---|---|---|---|
| | W-Q06→W-Q51 | W-Z02→W-Z35 | W-Z35→W-Z36 | W-S08→W-S16 | W-S16→W-S30 |
| $CaCO_3$ | 12.550 | 1.098 | −4.343 | −1.398 | −1.264 |
| $CaMg(CO_3)_2$ | 3.408 | −0.268 | 0.767 | 0.021 | −0.033 |
| $CaSO_4 \cdot 2H_2O$ | 3.246 | −0.109 | 3.693 | 1.476 | 2.284 |
| NaCl | 18.940 | −0.353 | 8.430 | 0.370 | −0.452 |
| $CO_2$ | −0.089 | 1.495 | 1.552 | 1.210 | 3.398 |
| $O_2$ | 20.170 | 0.159 | −4.624 | −2.696 | −4.478 |
| $H_2S$ | 10.080 | 0.079 | −2.312 | −1.348 | −2.239 |
| NaX | 41.080 | 1.833 | −0.648 | 0.167 | 2.038 |
| $CaX_2$ | −20.540 | −0.916 | 0.324 | −0.083 | −1.019 |

(Positive values indicate dissolution and negative values denote precipitation).

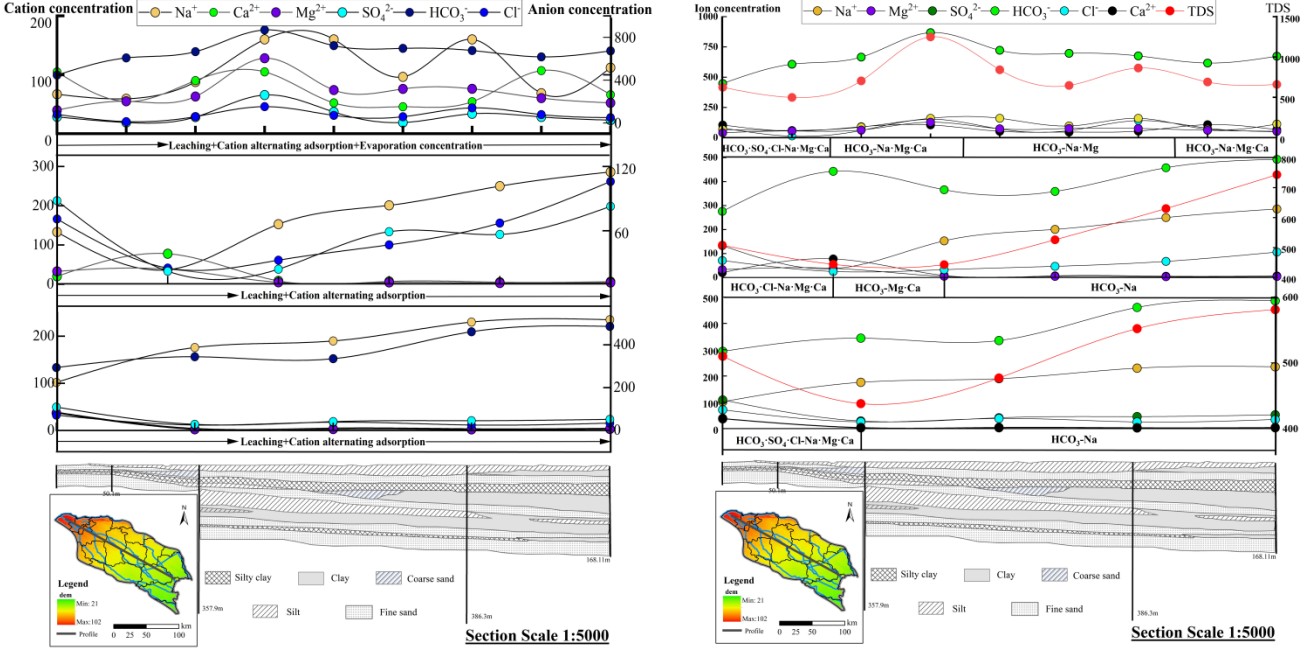

**Figure 7.** Hydrogeochemical evolution model of the three-layer groundwater aquifer along the chosen profile in the Guohe River Basin (unit: mg/L).

The results show that shallow groundwater is susceptible to atmospheric precipitation, human activities and evaporation and concentration, resulting in strong fluctuation of cation and anion concentrations therein. The concentrations of anions and cations in the intermediate layer are generally increasing, which is mainly caused by leaching and cation alternation adsorption, in a manner consistent with the reverse hydrogeochemical simulation results. $Na^+$ and $HCO_3^-$ in the deep layer augment significantly along the flow direction, indicating that the deep layer is affected by leaching and cation alternating adsorption.

According to the analysis in Section 3.1, the Guohe River Basin (Henan Section) has obvious zoning characteristics. The hydrochemical types of the shallow layer along the direction of groundwater flow vary, indicating that the shallow layer is affected by evaporation and concentration, and the intermediate and deep layers gradually change to $HCO_3$-Na type along the direction of groundwater flow (Figure 7). Leaching and cation alternating adsorption are the main hydrochemical processes affecting the three-layer aquifer in the basin.

## 4. Conclusions

The hydrochemical types of groundwater in this river basin (Henan Section) are as follows: an $HCO_3$-Na•Mg•Ca type predominates in the shallow layer, an $HCO_3$-Na type predominates in intermediate and deep layers and the groundwater has zoning characteristics. According to Gibbs diagrams, the lithogenesis of the three aquifers becomes more intense with increasing depth in the groundwater, and the weathering of the rock is gradually intensified. The shallow layer is affected by water–rock interaction, evaporation and concentration, while the intermediate and deep layers are mainly affected by the action of water on the rock.

The ion proportion coefficient method shows that $Na^+$ in groundwater is not completely derived from the dissolution of halite, and there may be dissolution of other sodium-bearing ores; $Ca^{2+}$ and $Mg^{2+}$ are mainly caused by the dissolution of evaporite and silicate minerals. The alternation adsorption of cations occurred in each aquifer, which reduced the content of $Ca^{2+}$ and $Mg^{2+}$ and increased the content of $Na^+$ in the solution.

PHREEQC shows that the SI index shows a general trend, such that SI > 0 and the groundwater is in a supersaturated state. Its shallow path is characterized by the dissolution of calcite, dolomite, gypsum and sulfur-bearing minerals. In the deep path, dolomite, gypsum and halite were dissolved, while calcite and sulfur minerals (pyrite, etc.) are precipitated. The deep path is characterized by the dissolution of gypsum and the precipitation of calcite, dolomite, halite, sulfur minerals and other minerals. The results are consistent with the calculated mineral SI data, indicating that water–rock interaction plays a major role in the geochemistry of these groundwater aquifers. Meanwhile, the NaX and $CaX_2$ dissolved precipitates of the three aquifers confirm quantitatively the cation-alternation adsorption between $Ca^{2+}$ and $Na^+$.

According to the hydrogeochemical evolution model, the dissolution and precipitation of mineral facies exert a significant influence on the evolution of groundwater chemistry. Along the direction of groundwater flow, the aquifers show an increasing trend of $HCO_3^-$ and $Na^+$ concentration, and in the vertical scale, the hydrochemical type is stable as $HCO_3$-Na along the direction of groundwater flow. The influence of cation alternation adsorption is significant, which verifies the trend in transformation of hydrochemistry type with increasing depth and qualitatively reveals the formation of the prevailing hydrogeochemistry in the Guohe River Basin (Henan Section). The ion concentrations in shallow groundwater fluctuate, and there is a small deviation from the results predicted by mineral simulation, which may be caused by human activities, evaporation and concentration.

**Author Contributions:** Conceptualization, F.Y.; methodology, F.Y.; investigation, F.Y. and D.Z.; data curation, X.L.; writing—original draft preparation, D.Z.; writing—review and editing, F.Y. and D.Z.; supervision, Z.L.; funding acquisition, F.Y. and Z.L. All authors have read and agreed to the published version of the manuscript.

**Funding:** This research was funded by the National Natural Science Foundation of China (Grant Nos 41402225 and 41972261), and the Key Scientific Research in Universities in Henan Province (Grant No. 21A170014).

**Data Availability Statement:** Not applicable.

**Conflicts of Interest:** The authors declare no conflict of interest.

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
