# Peer review of "Hydrochemical Characteristics and Hydrogeochemical Simulation Research of Groundwater in the Guohe River Basin (Henan Section)"

_water, doi:10.3390/w14091461_

Round 1

Reviewer 1 Report

The article presents hydrochemical research on the groundwater in Guohe River Basin. I propose making changes to the text. I hope my suggestions will improve the manuscript.

General  comments:

Authors should indicate elements of scientific novelty. As it stands, the article seems to deal with a local problem, but it may be of interest to a wider group of readers.

The environmental nature of the research requires a high standard of the analytical methods used. Authors should indicate and describe analytical procedures and validation of analytical procedures (e.eg information about basic metrological parameters –detection limits, precision, estimation of the uncertainty) should be added.

The number of significant figures (not decimal) should be according with the metrological rules based on the validation of the analytical method.

Author Response

  Thank you for the reviewers' comments concerning our manuscript entitled “Hydrochemical characteristics and hydrogeochemical simulation research of groundwater in the Guohe River Basin (Henan Section)”. Those comments are all valuable and very helpful for revising and improving our paper, as well as the important guiding significance to our researches. We have studied comments carefully and have made correction which we hope meet with approval.

  For details of the modifications, please refer to the annex.

Reviewer 2 Report

The role of the hydrosphere in the formation of the ecological state of the natural environment has always been significant, since the chemical composition of waters directly affects the physiological functions of a person and his health.

The manuscript of Yu Furong and co-authors is devoted to the study of hydrochemical characteristics of groundwater in the Guohe River Basin. The manuscript is well structured, the style of presentation is clear, the goals and objectives are obviously outlined.

However, there are a number of comments on the manuscript:

  1. Maybe add the word "groundwater" in the title of the article? “
  2. In the Introduction, the authors have perfectly outlined the history of the development of hydrogeochemistry in China, but, in our opinion, it would be nice to also provide references to hydrologists from other countries in this section.
  3. Unfortunately, Figures 1-4 and 6, 7 are not presented in very good quality in the manuscript. It would like their quality to be the same as that of Figure 5.
  4. In Figure 5, it is better to place the symbols below the diagrams.
  5. The number of sets should be entered in table 1, and not only mentioned in the text.
  6. The values of the standard deviations given in Table 1 are questionable.

Author Response

(The authors gave the same response as above.)

Round 2

Reviewer 1 Report

The article has been corrected as indicated by the reviewer and may be published in its current form.

Author Response

Dear Reviewers:

  Thank you for your letter and for the reviewers' comments concerning our manuscript entitled “Hydrochemical characteristics and hydrogeochemical simulation research of groundwater in the Guohe River Basin (Henan Section)”. According to the suggestion of the editor in chief, we have further revised the article. At the same time, in order to ensure that the main contents and main ideas of our article have not been changed, we will send you another copy of the revised manuscript.

  Finally, thank you again for your guidance and suggestions for this article, which has provided strong help for the further publication of this article.
